# What constrains food webs? A maximum entropy framework for predicting their structure with minimal biases

**Francis Banville**[1,2,3]*, **Dominique Gravel**[2,3], **Timothée Poisot**[1,3]

**1** Département de sciences biologiques, Université de Montréal, Montreal, Quebec, Canada, **2** Département de biologie, Université de Sherbrooke, Sherbrooke, Quebec, Canada, **3** Quebec Centre for Biodiversity Science, Quebec, Canada

* francis.banville@umontreal.ca

## Abstract

Food webs are complex ecological networks whose structure is both ecologically and statistically constrained, with many network properties being correlated with each other. Despite the recognition of these invariable relationships in food webs, the use of the principle of maximum entropy (MaxEnt) in network ecology is still rare. This is surprising considering that MaxEnt is a statistical tool precisely designed for understanding and predicting many types of constrained systems. This principle asserts that the least-biased probability distribution of a system's property, constrained by prior knowledge about that system, is the one with maximum information entropy. MaxEnt has been proven useful in many ecological modeling problems, but its application in food webs and other ecological networks is limited. Here we show how MaxEnt can be used to derive many food-web properties both analytically and heuristically. First, we show how the joint degree distribution (the joint probability distribution of the numbers of prey and predators for each species in the network) can be derived analytically using the number of species and the number of interactions in food webs. Second, we present a heuristic and flexible approach of finding a network's adjacency matrix (the network's representation in matrix format) based on simulated annealing and SVD entropy. We built two heuristic models using the connectance and the joint degree sequence as statistical constraints, respectively. We compared both models' predictions against corresponding null and neutral models commonly used in network ecology using open access data of terrestrial and aquatic food webs sampled globally (N = 257). We found that the heuristic model constrained by the joint degree sequence was a good predictor of many measures of food-web structure, especially the nestedness and motifs distribution. Specifically, our results suggest that the structure of terrestrial and aquatic food webs is mainly driven by their joint degree distribution.

## Author summary

Predator and prey species form complex networks of energy flow that drive the functioning and dynamics of ecosystems. Because of their high number and variability, sampling and documenting these feeding relationships are particularly challenging, especially at

**Data Availability Statement:** All code and data to reproduce this article are available at the Open Science Framework (https://osf.io/kt4gs/files/github).

**Funding:** This work was supported by the Institute for Data Valorization (IVADO, PhD-2019a-5993304626), which supported FB, and the Natural Sciences and Engineering Research Council of Canada (NSERC) Collaborative Research and Training Experience (CREATE) program, through the Computational Biodiversity Science and Services (BIOS2) program, which supported FB, DG, and TP. TP received funding from the Canadian Institute for Ecology & Evolution and a donation from the Courtois Foundation. TP also acknowledges financial support from NSERC through the Discovery Grants and Discovery Accelerator Supplement programs. The funders had no role in study design, data collection and analysis, decision to publish, or preparation of the manuscript.

**Competing interests:** The authors have declared that no competing interests exist.

large spatial and temporal scales. Fortunately, many properties of food webs can be studied without knowing every trophic interaction in a network by using suitable computational methods. This is notably the case of their emerging structure, which could be driven by just a few ecological and biological variables despite the large number of mechanisms shaping ecological networks. In this contribution, we present a novel framework to identify these variables by applying the principle of maximum entropy to the analysis of food-web structure. We show that the number of prey and predators for each species in a food web is a fundamental property that shapes many aspects of the network, while also being predictable by our model and available data. Our approach, which can be applied to a wide range of biological and ecological networks, can be used to make better predictions of food webs across space, in addition to contributing to our understanding of the ecological processes shaping predatory interactions at the community level.

## Introduction

### The constrained structure of ecological networks

A variety of measures of the structure of ecological networks have been used to describe the organization of species interactions in a biological community [1]. These measures provide valuable information on the functioning of ecosystems and their responses to environmental change (e.g. [2, 3]). For instance, two main properties of a network are its order (the number of nodes or species $S$) and its size (the number of edges or species interactions $L$), whose ratio $L/S^2$ (the proportion of possible interactions that are realized) is called connectance. In food webs, [4] showed that a high connectance can promote the robustness of the community to species lost. Moreover, a different investigation of network structure revealed that plant–pollinator and seed-disperser networks have a highly nested structure that can promote species persistence [5]. Despite the recognition of the ecological significance of network structure, our knowledge of the properties of ecological networks remains limited and is unevenly distributed across space [6]. This can largely be attributed to the difficulties encountered when sampling interactions [7], which impede our ability to describe network structure on a global scale. Moreover, these complex systems [8] have an emerging structure with intricate dynamics (i.e. non-linear behaviors that cannot be predicted from individual interactions). Their multi-faceted nature and high level of complexity represent a challenge when studying the ecological processes driving their structure, especially given the strong relationship between many of their properties. For example, nestedness and modularity are highly correlated in ecological networks [9] and many emerging properties are linked to connectance [10]. In light of these observations, it is difficult to assess whether the presumed effects of a particular measure on network structure and behavior are the artifacts of other, perhaps simpler, measures.

One way to address this issue is to recognize that food webs and other ecological networks are constrained systems. In other words, the space of possible network configurations is shaped by our partial knowledge of their structure. For example, there is a finite set of networks with a given number of species and interactions (i.e. a given connectance). As shown by [10], connectance can constrain many aspects of network structure such as the degree distribution (i.e. the probability distribution of the number of interactions realized by a species). Because of the scaling relationship between the total number of species and interactions, network structure is also largely constrained by its order (food webs with high species richness typically have a low connectance compared to smaller networks, [11]). Other measures, such as the maximum trophic level (i.e. the maximum number of times energy is transformed along

food chains through biomass consumption), may also constrain the space of feasible networks, e.g. by imposing a limit on energy transfer [12].

Prior knowledge of the structure of ecological networks, even partial, can be useful in the current context of data scarcity about species interactions. The Eltonian shortfall, which describes the gap between our present-day knowledge of ecological networks and a comprehensive understanding of interactions [13], can be partially alleviated using known information about a network's attributes. First, knowing a given set of network properties (e.g. the number of species and interactions), it is possible to identify various structures that meet these constraints and estimate their respective probabilities of occurrence. For instance, a set of adjacency matrices (i.e. a representation of interactions in matrix format) that satisfy the known properties can be found. Selecting the right adjacency matrix among all of these suitable configurations or evaluating their relative probabilities may be done using different modeling tools (e.g. using the principle of maximum entropy or Bayesian inference). Second, as suggested by [14], network structure can also be used to facilitate the estimation of pairwise species interactions. For example, by knowing a network's adjacency matrix (or the constrained space of feasible networks), we can decrease the volume of information we need to infer interactions (e.g. by knowing how many prey each species should have). Moreover, network structure can be used as validation to ensure that inferred interactions collectively satisfy known structural properties. Overall, partial knowledge of a network's properties can help us reconstruct its emerging structure and individual interactions when data is lacking.

Predicting ecological networks and understanding the biological mechanisms that shape species interactions are two complementary aims of network ecology. On one hand, prediction is the process of estimating the value of an unknown variable using available data and appropriate statistical models and assumptions. The goal of predictive models is to minimize predictive errors and provide reliable estimates of the unknown variable. These models have the potential to fill many data gaps about species interactions and network structure by making sound predictions and generating new data. A variety of such models have recently been developed using machine learning and other statistical tools, most of which are presented in [14]. On the other hand, statistical models can also be built to describe or understand a parameter or mechanism of interest. For example, null models are statistical models that generate baseline distributions to evaluate the significance of an observed measure. These distributions are obtained from different randomizations of a network that maintain some of its properties while deliberately omitting others. By assessing the deviation of empirical data from these reference distributions, we can evaluate the degree to which the processes underlying the null model can accurately capture the observed measures, providing a benchmark to assess the importance of additional factors such as the omitted properties [1, 15]. This makes null models a valuable tool to identify the ecological mechanisms that drive species interactions and constrain the structure of ecological networks. The distinction between understanding and predicting is crucial when using statistical and mathematical models as this can impact how we use and interpret them. Numerous predictive and null models have been used to analyze ecological networks [1, 14]. Nevertheless, given the constrained nature of ecological networks, it is surprising that the principle of maximum entropy, a mathematical method designed for both the description and prediction of constrained systems, has been barely used in network ecology.

## The principle of maximum entropy: A primer for ecologists

The principle of maximum entropy (MaxEnt) is a mathematical method of finding probability distributions, strongly rooted in statistical mechanics and information theory [16–18]. Starting

from a set of constraints given by prior knowledge of a system (i.e. what we call state variables), this method helps us find least-biased probability distributions subject to the constraints. These probability distributions are guaranteed to be unique given our prior knowledge and represent the most we can say about a system without making more assumptions. For example, if we know the number of species and total number of individuals in a biological community (our state variables), we can calculate the average number of individuals per species (our constraint). Using the principle of maximum entropy, one could show that the least-biased species abundance distribution (i.e. the distribution of the number of individuals of each species), constrained by the average species abundance, follows an exponential distribution whose parameter is determined by the values of the state variables [19, 20]. However, this does not imply that this distribution will be the best fit to empirical data. The challenge is to find the right set of constraints that would best reproduce distributions found in nature.

MaxEnt states that the least-biased probability distribution given the constraints used is the one with the highest entropy (a measure of the uncertainty of a random process) among all probability distributions that satisfy these constraints. Many measures of entropy have been developed in physics [21], but only a fraction of them could be used as an optimization measure with the principle of maximum entropy. According to [21] and [22], a measure of entropy $H$ should satisfy four properties: (1) it should be a function of a probability distribution $p(n)$ only; (2) it should be maximized when $p(n)$ is uniform; (3) it should not be influenced by outcomes with a null probability; and (4) it should be independent of the order of information acquisition. The Shannon's entropy [23]

$$H = -\sum_n p(n) \log p(n) \tag{1}$$

satisfies all of these properties. Finding the probability distribution $p(n)$ that maximizes $H$ under a set of $m$ constraints $g$ can be done using the method of Lagrange multipliers. These constraints can include one or many properties of the probability distribution (e.g. its mean, variance, and range). However, the normalization constraint always needs to be included in $g$ to make sure that $p(n)$ sums to 1. The objective is then to find the values of the Lagrange multipliers $\lambda_i$ that optimize a function $F$:

$$F = H - \sum_{i=1}^{m} \lambda_i(g_i - c_i), \tag{2}$$

where $g_i$ is the mathematical formulation of the constraint $i$ and $c_i$, its value. Note that $F$ is just Shannon's entropy to which we added terms that each sums to zero ($g_i = c_i$). $F$ is maximized by setting to 0 its partial derivative with respect to $p(n)$.

The principle of maximum entropy has been used in a wide range of disciplines, from thermodynamics, chemistry and biology [24] to graph and network theory (e.g. [25, 26]). MaxEnt has also been proven useful in ecology, e.g. in species distribution [27] and macroecological [20, 28] models. In network ecology, it can be used to generate null models of network structure with either fixed or fluctuating constraints [29] and predict important network properties using limited data. For example, it has been used to predict the degree distribution of bipartite networks from the number of species and the number of interactions [30] and the strength of biotic interactions from species relative abundances [31]. However, to the best of our knowledge, MaxEnt has never been used to predict food-web structure directly, even though food webs are among the most documented and widespread ecological networks [32].

Food-web properties that can be derived using MaxEnt are varied and pertain to different elements of the network (i.e. at the species, interaction and community levels). Because

MaxEnt is a method of finding least-biased probability distributions given partial knowledge about a system, these properties need to be represented probabilistically. For example, at the species level, MaxEnt can be used to predict the distribution of trophic levels among species, as well as the distribution of species' vulnerability (number of predators) and generality (number of prey). By contrast, at the interaction level, predictions can be made on the distribution of interaction strengths in weighted food webs. At the community level, it can generate probability distributions of many measures of a network's emerging structure and of networks themselves (i.e. a probability distribution governing the occurrence of different network configurations). Because the decomposition of a network's adjacency matrix into a product of matrices can yield a vector of relative values (e.g. a vector of relative singular values), MaxEnt can also be used to find the configuration whose distribution of relative values is of maximum entropy. This configuration would be the one with the greatest entropy (or complexity, [33]) among all configurations that satisfy the constraints. Overall, the potential of this method in the study of food webs is broad. The applicability and performance of MaxEnt mostly depend on the ecological information available and on our capacity to find the right set of state variables that best represent natural systems and to translate them into appropriate statistical constraints. Having a validated maximum entropy model for the system at hand allows us to make least-biased predictions using a minimal amount of data, as well as identify the most important ecological processes shaping that system. In other words, MaxEnt can help us better understand and predict the structure of ecological networks worldwide.

## Analytical and heuristic approaches

In this contribution, we used two complementary approaches to predict the structure of food webs using the principle of maximum entropy. The first approach consists in deriving constrained probability distributions of given network properties analytically, whereas the second approach consists in finding the adjacency matrix of maximum entropy heuristically, from which network properties can be measured. We compared our predictions against empirical data and null and neutral models commonly used in network ecology. We focus on deterministic (non-probabilistic) and unweighted (Boolean) food webs in both approaches for two reasons. (1) Most sampled food webs lack estimates of interaction probabilities (e.g. spatial variability of interactions) and strengths (e.g. energy fluxes between species) because of inherent difficulties in measurement. (2) The methods and measures to analyze the structure of binary food webs are more developed and tested [1], which increases the range of analysis we can perform and simplifies their interpretation. However, our framework can be applied to all types of ecological networks and a wider variety of measures.

For the first approach (analytic), we focus on species-level properties. Specifically, we derived the joint degree distribution (i.e. the joint probability distribution that a species has a given number of prey and predators in its network) of maximum entropy using only the number of species $S$ and the number of interactions $L$ as state variables. Then, we calculated the degree distribution of maximum entropy directly from the joint degree distribution by summing probabilities over the joint degree distribution (since the degree of a species is its total number of prey and predators combined). Because of the scarcity of empirical data on the number of interactions in food webs, we also present a method to predict $L$ from $S$ (Box 1), thus allowing the prediction of the joint degree distribution from $S$ solely.

For the second approach (heuristic), we focus on community-level properties. We used a flexible and heuristic model based on simulated annealing (an optimization algorithm, [34]) to find the network configuration *close* to maximum entropy and measured its structure. We developed this heuristic model because the analytical derivation of a maximum entropy graph

model of food webs is difficult, and because this model is readily applicable to other types of ecological networks and measures. Indeed, the mathematical representation of food webs (i.e. directed simple graphs frequently having self-loops) makes the optimization of maximum entropy graph models more complicated than with many other types of (non-ecological) networks. In other words, deriving a probability distribution on the graphs themselves is difficult when working with food webs. We built two types of heuristic MaxEnt models depending on the constraint used. Our type I MaxEnt model uses the connectance of the network (i.e. the ratio $L/S^2$) as a constraint, whereas our type II MaxEnt model uses the whole joint degree sequence as a constraint.

## Methods

### Analytical maximum entropy models

The analytical approach is the most common way to use and develop maximum entropy models. As shown above, starting from a defined set of constraints, a mathematical expression for the target distribution is derived using the method of Lagrange multipliers. The derived distribution of maximum entropy is unique and least biased given the constraints used. Although we refer to this approach as analytic, finding the values of the Lagrange multipliers usually requires the use of numerical methods. Here we use MaxEnt to derive two species-level properties in food webs: the joint degree distribution and the degree distribution. The degree distribution has driven the attention of ecologists because of its role in determining the assembly of ecological networks [35], shaping their emerging structure [9], and understanding interaction mechanisms [30]. As noted above, although the degree distribution of maximum entropy has already been derived in bipartite networks [30], we show in much greater detail its mathematical derivation in food webs. But first, we derived the joint degree distribution, a related property that holds significantly more ecological information than the degree distribution.

We tested our analytical MaxEnt models against open food-web data queried from three different sources and integrated into what we call our *complete dataset*. These sources include (1) terrestrial and aquatic food webs sampled globally and archived on the ecological interactions database Mangal; (2) freshwater stream food webs from New Zealand; and (3) aquatic food webs from Tuesday Lake (Michigan, United States). First, all food webs archived on mangal.io [36, 37] were directly queried from the database ($N = 235$). Most ecological networks archived on Mangal are multilayer networks, i.e. networks that describe different types of interactions. We kept all networks whose interactions were mainly of predation and herbivory types and removed the largest network ($S = 714$) for computational efficiency reasons. Then, to this set, we added food webs from two different sources: the New Zealand dataset ($N = 21$, [38]) and the Tuesday Lake dataset ($N = 2$, [39]). Our complete dataset thus contained a total of 257 food webs. These complex food webs differ in their level of resolution and sampling effort, which may introduce noise in the estimation of their properties, especially given their large number of interacting elements. However, because our MaxEnt models are applied on imperfect data, they aim at reproducing the *sampled* structure of food webs, not their actual structure. All code and data to reproduce this article are available at the Open Science Framework (https://osf.io/kt4gs/files/github). Data cleaning, simulations and analyses were conducted in Julia v1.8.0.

**Joint degree distribution.** The joint degree distribution $p(k_{in}, k_{out})$ of a food web with $S$ species is a joint discrete probability distribution describing the probability that a species has $k_{in}$ predators and $k_{out}$ prey, with $k_{in}$ and $k_{out} \in [0, S]$. Basal species (e.g. plants) have a $k_{out}$ of 0, whereas top predators have a $k_{in}$ of 0. In contrast, the maximum number of prey and predators

a species can have is set by the total number of species in the food web. Here we show how the joint degree distribution of maximum entropy can be obtained given knowledge of the number of species $S$ and the number of interactions $L$.

We want to maximize Shannon's entropy

$$H = -\sum_{k_{in}=0}^{S} \sum_{k_{out}=0}^{S} p(k_{in}, k_{out}) \log p(k_{in}, k_{out}) \tag{3}$$

subject to the following constraints:

$$g_1 = \sum_{k_{in}=0}^{S} \sum_{k_{out}=0}^{S} p(k_{in}, k_{out}) = 1; \tag{4}$$

$$g_2 = \sum_{k_{in}=0}^{S} \sum_{k_{out}=0}^{S} k_{in} p(k_{in}, k_{out}) = \langle k_{in} \rangle = \frac{L}{S}; \tag{5}$$

$$g_3 = \sum_{k_{in}=0}^{S} \sum_{k_{out}=0}^{S} k_{out} p(k_{in}, k_{out}) = \langle k_{out} \rangle = \frac{L}{S}. \tag{6}$$

The first constraint $g_1$ is our normalizing constraint, whereas the other two ($g_2$ and $g_3$) fix the average of the marginal distributions of $k_{in}$ and $k_{out}$ to the linkage density $L/S$. It is important to notice that $\langle k_{in} \rangle = \langle k_{out} \rangle$ because every edge is associated with a predator and a prey. Therefore, without using any further constraints, we would expect the joint degree distribution of maximum entropy to be a symmetric probability distribution with regards to $k_{in}$ and $k_{out}$. However, this does not mean that the joint degree *sequence* will be symmetric since the joint degree sequence is a random realization of its probabilistic counterpart.

The joint probability distribution of maximum entropy given these constraints is found using the method of Lagrange multipliers. To do so, we seek to maximize the following expression:

$$F = H - \lambda_1(g_1 - 1) - \lambda_2\left(g_2 - \frac{L}{S}\right) - \lambda_3\left(g_3 - \frac{L}{S}\right), \tag{7}$$

where $\lambda_1$, $\lambda_2$, and $\lambda_3$ are the Lagrange multipliers. The probability distribution that maximizes entropy is obtained by finding these values. As pointed out above, $F$ is just Shannon's entropy to which we added terms that each sums to zero (our constraints). $F$ is maximized by setting to 0 its partial derivative with respect to $p(k_{in}, k_{out})$. Because the derivative of a constant is zero, this gives us:

$$\frac{\partial H}{\partial p(k_{in}, k_{out})} = \lambda_1 \frac{\partial g_1}{\partial p(k_{in}, k_{out})} + \lambda_2 \frac{\partial g_2}{\partial p(k_{in}, k_{out})} + \lambda_3 \frac{\partial g_3}{\partial p(k_{in}, k_{out})}. \tag{8}$$

Evaluating the partial derivatives with respect to $p(k_{in}, k_{out})$, we obtain:

$$-\log p(k_{in}, k_{out}) - 1 = \lambda_1 + \lambda_2 k_{in} + \lambda_3 k_{out}. \tag{9}$$

Then, solving Eq (9) for $p(k_{in}, k_{out})$, we obtain:

$$p(k_{in}, k_{out}) = \frac{e^{-\lambda_2 k_{in} - \lambda_3 k_{out}}}{Z}, \tag{10}$$

where $Z = e^{1+\lambda_1}$ is called the partition function. The partition function ensures that

probabilities sum to 1 (our normalization constraint). It can be expressed in terms of $\lambda_2$ and $\lambda_3$ as follows:

$$Z = \sum_{k_{in}=0}^{S} \sum_{k_{out}=0}^{S} e^{-\lambda_2 k_{in} - \lambda_3 k_{out}}. \tag{11}$$

After substituting $p(k_{in}, k_{out})$ in Eqs (5) and (6), we get a nonlinear system of two equations and two unknowns:

$$\frac{1}{Z} \sum_{k_{in}=0}^{S} \sum_{k_{out}=0}^{S} k_{in} e^{-\lambda_2 k_{in} - \lambda_3 k_{out}} = \frac{L}{S}; \tag{12}$$

$$\frac{1}{Z} \sum_{k_{in}=0}^{S} \sum_{k_{out}=0}^{S} k_{out} e^{-\lambda_2 k_{in} - \lambda_3 k_{out}} = \frac{L}{S}. \tag{13}$$

We solved Eqs (12) and (13) numerically using the Julia library JuMP.jl v0.21.8 [40]. JuMP. jl supports nonlinear optimization problems by providing exact second derivatives that increase the accuracy and performance of its solvers. The estimated values of $\lambda_2$ and $\lambda_3$ can be substituted in Eq (10) to have a more workable expression for the joint degree distribution.

We assessed the empirical support of this expression using all food webs in our complete dataset. First, we predicted the joint degree distribution of maximum entropy for each of these food webs, i.e. using their number of species and number of interactions as state variables. Then, we sampled one realization of the joint degree sequence for each network using the probabilities given by the joint degree distribution, while fixing the total number of interactions. This gave us a random realization of the number of prey and predators for each node in each network. We standardized the predicted $k_{out}$ and $k_{in}$ by the total number of species in their network to generate relative values, which can be compared across networks.

**Degree distribution.**   The joint degree distribution derived using MaxEnt (Eq (10)) can be used to obtain the degree distribution of maximum entropy. The degree distribution $p(k)$ represents the probability that a species has $k$ interactions in its food web, with $k = k_{in} + k_{out}$. It can thus be obtained from the joint degree distribution as follows:

$$p(k) = \sum_{i=0}^{k} p(k_{in} = k - i, k_{out} = i). \tag{14}$$

The degree distribution can also be obtained directly using the principle of maximum entropy, as discussed in [30]. This gives the following distribution:

$$p(k) = \frac{e^{-\lambda_2 k}}{Z}, \tag{15}$$

with $Z = \sum_{k=0}^{S} e^{-\lambda_2 k}$.

This can be solved numerically using the constraint of the average degree $\langle k \rangle = \frac{2L}{S}$ of a species, yielding an identical solution to the one obtained using the joint degree distribution as an intermediate.

$$\frac{1}{Z} \sum_{k=0}^{S} k e^{-\lambda_2 k} = \frac{2L}{S} \tag{16}$$

Note that the mean degree is twice the value of the linkage density because every interaction must be counted twice when we add in and out-degrees together.

## Heuristic maximum entropy models

With the analytical approach, we showed how important measures of food-web structure (e.g. the degree distribution and the joint degree distribution) can be derived with the principle of maximum entropy using minimal knowledge about a biological community. This type of model, although useful to make least-biased predictions of many network properties, can be hard to apply for other measures. There are dozens of measures of network structure [1] and many are not directly calculated with mathematical equations but are instead estimated algorithmically. Moreover, the applicability of this method to empirical systems is limited by the state variables we can measure and use. Here, we suggest a more flexible method to predict many measures of network structure simultaneously, i.e. by finding heuristically the network configuration of maximum SVD (singular value decomposition) entropy given partial knowledge of its emerging structure.

**From Shannon's to SVD entropy.**   The principle of maximum entropy can be applied on the network itself if we decompose its adjacency matrix into a non-zero vector of relative values. This is a necessary step when working with food webs, which are frequently expressed as a matrix $A = [a_{ij}]$ of Boolean values representing the presence ($a_{ij} = 1$) or absence ($a_{ij} = 0$) of an interaction between two species $i$ and $j$. Knowing one or many properties of a food web of interest (e.g. its number of species and number of interactions), we can simulate its adjacency matrix randomly by using this known ecological information to constrain the space of potential networks. The entropy of this hypothetical matrix can then be measured after decomposing it into appropriate values. Simulating a series of networks until we find the one having the highest entropy allows us to search for the most complex food-web configuration given the ecological constraints used. This configuration is the least-biased one considering the information available. In other words, the most we can say about a network's adjacency matrix, without making more assumptions than the ones given by our incomplete knowledge of its structure, is the one of maximum entropy. Generating the most complex network that corresponds to this structure allows us to explore more easily other properties of food webs under MaxEnt.

Shannon's entropy can only be calculated on conventional probability distributions such as the joint degree distribution. This is an issue when working with the adjacency matrix of ecological networks. For this reason, we need to use another measure of entropy if we want to predict a network's configuration directly using MaxEnt. We used the SVD entropy as our measure of entropy, which is an application of Shannon's entropy to the relative non-zero singular values of a food web's adjacency matrix [33]. These values are obtained by performing singular value decomposition (SVD) on the matrix. SVD is a mathematical method that decomposes a matrix into three separate matrices, one of which contains singular values on the diagonal. We selected the non-zero singular values (i.e. the informative singular values that reflect a non-negligible contribution to the structure of the matrix) by keeping the largest $R$ values, where $R$ is the rank of the matrix [41]. We measured SVD entropy as follows:

$$J = -\sum_{i=1}^{R} s_i \log s_i, \tag{17}$$

where $s_i$ are the relative singular values of the adjacency matrix ($s_i = \sigma_i / \sum_{i=1}^{R} \sigma_i$, where $\sigma_i$ are the singular values). Note that the distribution of relative singular values is analogous to a probability distribution, with $0 < s_i < 1$ and $\Sigma s_i = 1$. This measure also satisfies all four

properties of an appropriate entropy measure above-mentioned, while being a proper measure of the internal complexity of food webs [33]. Following [33], we standardized it with the rank of the matrix (i.e. $J/\ln(R)$) to account for the difference in dimensions between networks (*sensu* Pielou's evenness, [42]).

**Types I and II heuristic MaxEnt models.** We used SVD entropy to predict the network configuration of maximum entropy (i.e. of maximum complexity) heuristically given different constraints for all food webs in our complete dataset. We built two types of heuristic MaxEnt models that differ on the constraint used. The type I heuristic MaxEnt model is based on connectance, whereas the type II heuristic MaxEnt model is based on the joint degree sequence. These models are thus based on the same constraints as the types I [15] and II [5] null models (Box 2) frequently used to generate random networks topologically. This allows a direct comparison of the performance of null and heuristic MaxEnt models in reproducing the emerging structure of empirical food webs.

For each network, we estimated their configuration of maximum entropy given each of these constraints. For both types of heuristic MaxEnt models, we used a simulated annealing algorithm with 4 chains, 2000 steps and an initial temperature of 0.2. For each food web, we first generated one random Boolean matrix per chain while fixing the number of species. We also maintained the total number of interactions (i.e. the sum of all elements in the matrix) in the type I MaxEnt model and the joint degree sequence (i.e. the rows and columns sums) in the type II MaxEnt model. These were our initial configurations. Then, we swapped interactions sequentially while maintaining the original connectance or joint degree sequence. Configurations with a higher SVD entropy than the previous one in the chain were always accepted, whereas they were accepted with a probability conditional to a decreasing temperature and the difference in SVD entropy when lower. The final configuration with the highest SVD entropy among the four chains constitutes the estimated maximum entropy configuration of a food web given the constraint used.

**Structure of MaxEnt food webs.** We measured various properties of these configurations of maximum entropy to evaluate how well they reproduce the structure of sampled food webs. Specifically, we evaluated their nestedness $\rho$, their maximum trophic level *maxtl*, their network diameter *diam*, their average maximum similarity between species pairs *MxSim* [8], their proportion of cannibal species *Cannib*, their proportion of omnivorous species *Omniv*, their SVD entropy, and their motif profile. Nestedness indicates how much the diet of specialist species is a subset of the one of generalists [1] and was measured using the spectral radius of the adjacency matrix [43]. Next, the network diameter represents the longest of the shortest paths between all species pair [44]. Further, cannibal species are species that can eat individuals of their own species (i.e. species having self-loops), whereas omnivorous species can prey on different trophic levels [8]. Finally, motifs are unique n-species connected subgraphs that can be considered simple building blocks of ecological networks [45, 46]. There are 13 possible three-species motifs in food webs, including 5 with only single links ($i \rightarrow j$) and 8 with double links ($i \leftrightarrow j$), labeled S1-S5 and D1-D8 by [46], respectively. A motif profile represents the proportion of each of these motifs in a network (i.e. their relative frequencies). All of these properties are relatively easy to measure and were chosen based on their ecological importance and prevalent use in network ecology [1, 47].

We compared the performance of both heuristic MaxEnt models in predicting these measures to the one of the null and neutral models (Box 2). We conducted these comparisons using two different datasets: (1) our complete dataset including most food webs archived on Mangal, as well as all food webs in the New Zealand and Tuesday Lake datasets, and (2) our *abundance dataset*, a subset of the complete dataset comprising all food webs having data on their species' relative abundances ($N = 19$). Indeed, of the New Zealand and Tuesday Lake

datasets, 19 networks had data on species' relative abundances that were used in the neutral model to better assess the performance of our heuristic models. We compared our models' predictions using these two datasets separately to minimize biases and to better represent food webs with abundance data.

## Box 1: Working with predicted numbers of interactions

Our analytical MaxEnt models require information on the number of species and the number of interactions. However, since the latter is rarely measured empirically, ecologists might need to use another predictive model to estimate the total number of interactions in a food web before using MaxEnt. Here we show how this can be done by combining both models sequentially.

We used the flexible links model of [11] to predict the number of interactions from the number of species in food webs. The flexible links model has been shown to make reliable predictions of $L$ while taking into account meaningful ecological constraints on its lower and upper bounds. Specifically, recognizing that there is a minimum of $S - 1$ (if no isolated species) and a maximum of $S^2$ (if all species interact) interactions in food webs, it estimates the number of the $S^2 - (S - 1)$ *flexible links* that are realized. This represents the number of realized interactions above the minimum (i.e. $L_{FL} = L - (S - 1)$). The probability that a flexible link is realized is treated as constant within a particular food web but differs across food webs to capture variations in connectance. Therefore, assuming that flexible links are independent Bernoulli events, the total number of flexible links that are realized in a food web can be obtained from a beta-binomial distribution BB with $S^2 - (S - 1)$ trials and parameters $\alpha = \mu e^\phi$ and $\beta = (1 - \mu)e^\phi$:

$$L_{FL} \sim \mathrm{BB}(S^2 - (S - 1), \mu e^\phi, (1 - \mu)e^\phi), \tag{18}$$

where $\mu$ is the average probability across food webs that a flexible link is realized and $\phi$ is the concentration parameter around $\mu$.

We fitted the flexible links model on all food webs in our complete dataset and estimated the parameters of Eq (18) using a Hamiltonian Monte Carlo sampler with static trajectory (4 chains and 3000 iterations):

$$[\mu, \phi | \mathbf{L}, \mathbf{S}] \propto \prod_{i=1}^{m} \mathrm{BB}(L_i - (S_i - 1) | S_i^2 - (S_i - 1), \mu e^\phi, (1 - \mu)e^\phi)$$
$$\times \mathrm{B}(\mu | 3, 7) \times \mathcal{N}(\phi | 3, 0.5), \tag{19}$$

where $m$ is the number of food webs ($m = 257$) and $\mathbf{L}$ and $\mathbf{S}$ are the vectors of their numbers of interactions and numbers of species, respectively. Our weakly-informative prior distributions were chosen following [11], i.e. a beta distribution for $\mu$ and a normal distribution for $\phi$. The Monte Carlo sampling of the posterior distribution was conducted using the Julia library Turing v0.15.12.

The flexible links model is a generative model, i.e. it can generate plausible values of the predicted variable. We thus simulated 1000 values of $L$ for different values of $S$ using the joint posterior distribution of our model parameters (Eq (19)) and calculated the mean degree for each simulated value. The resulting distributions are shown in the left panel of Fig 1 for three different values of species richness. In the right panel of Fig 1, we show how the probability distribution for the mean degree constraints can be used to generate a distribution of maximum entropy degree distributions since each

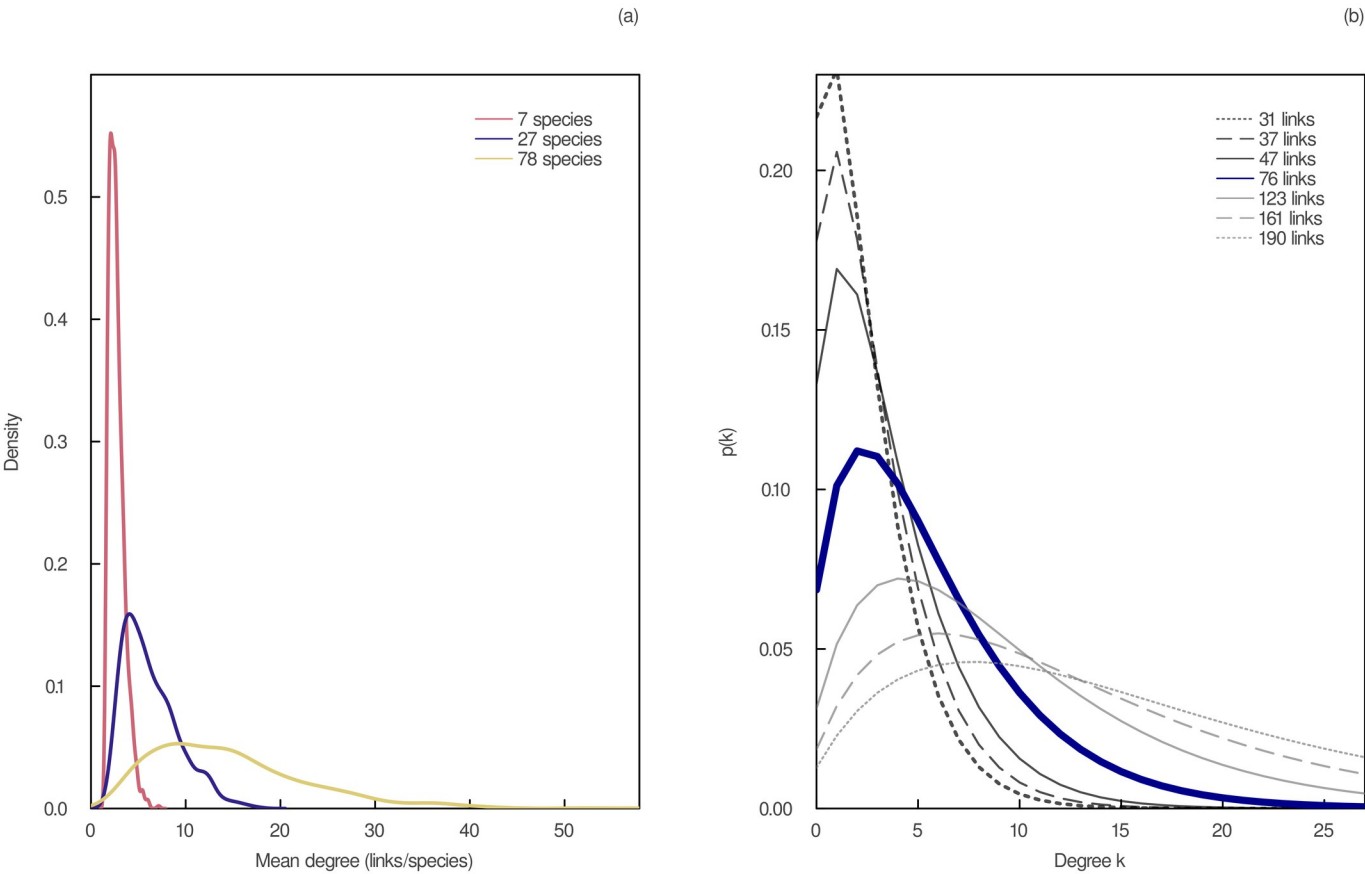

**Fig 1. Maximum entropy degree distributions with predicted numbers of interactions.** (a) Probability density of the mean degree of a food web obtained using different values of species richness *S*. The number of interactions *L* was simulated 1000 times using the flexible links model fitted to all empirical networks (Eq (19)). The mean degrees 2*L*/*S* were then obtained from these simulated values. (b) Degree distributions of maximum entropy for a network of *S* = 27 species and different numbers of interactions. The numbers of interactions correspond to the lower and upper bounds of the 67%, 89%, and 97% percentile intervals (PI), as well as the median (in blue), of the counterfactuals of the flexible links model. Each degree distribution of maximum entropy was obtained using Eq (15) after solving numerically Eq (16) using different values of the mean degree constraint.

simulated value of mean degree generates a different maximum entropy degree distribution (Eqs (15) and (16)).

## Box 2: Corresponding null and neutral models

### Null models (types I and II)

The predictions of our heuristic maximum entropy models were compared against two topological null models. These null models use the same ecological information as our heuristic models and thus constitute an adequate baseline for comparison. The first is the type I null model of [15], in which the probability that a species *i* predates on another species *j* is given by

$$p(i \rightarrow j) = \frac{L}{S^2}. \tag{20}$$

The second is the type II null model of [5], in which the probability of interaction is instead given by

$$p(i \rightarrow j) = \frac{1}{2}\left(\frac{k_{in}(j)}{S} + \frac{k_{out}(i)}{S}\right),$$ (21)

where $k_{in}(j)$ and $k_{out}(i)$ are the in and out-degrees of species $j$ and $i$, respectively. The type I null model is based on connectance, whereas the type II null model is based on the joint degree sequence. Therefore, the type I and II topological null models correspond to our type I and II heuristic MaxEnt models, respectively, since they use similar constraints.

We generated probabilistic networks using both types of null models for all empirical food webs in our complete dataset. Then, we converted these networks to adjacency matrices of Boolean values by generating 100 random networks for each of these probabilistic webs and kept the $L$ entries that were sampled the most amount of times, with $L$ given by the number of interactions in each food web. This ensured that the resulting null networks had the same number of interactions as their empirical counterparts. Thus, for each null model, we ended up with one null adjacency matrix for each empirical network.

### Neutral model

We also compared our heuristic MaxEnt models to a neutral model of relative abundances, in which the probabilities of interaction are given by

$$p(i \rightarrow j) \propto \frac{n_i}{N} \times \frac{n_j}{N},$$ (22)

where $n_i$ and $n_j$ are the abundances (or biomass) of both species and $N$ is the total abundance (or biomass) of all species in the network. We generated neutral abundance matrices for all empirical food webs in our abundance dataset and converted these weighted networks to adjacency matrices of Boolean values using the same method as the one we used for our null models.

## Results and discussion

### Analytical maximum entropy models

We first discuss the predictive capacity of our analytical models. The relationship between the relative numbers of prey $k_{out}$ and predators $k_{in}$ in empirical networks and obtained from the joint degree distributions of maximum entropy is depicted in the left and central panels of Fig 2, respectively. We observe that our analytical model predicts higher values of generality and vulnerability compared to empirical food webs (i.e. relative values of $k_{out}$ and $k_{in}$ both closer to 1) for many species. In other words, our model predicts that species that have many predators also have more prey than what is observed empirically (and conversely). This is not surprising, given that our model did not include biological factors preventing generalist predators from having many prey. Nevertheless, with the exception of these generalist species, MaxEnt adequately predicts that most species have low generality and vulnerability values.

Examining the difference between predicted and empirical values for each species gives a slightly different perspective (right panel of Fig 2). To make that comparison, we must first

associate each of our predictions with a specific species in a network. Indeed, our predicted joint degree sequences have the same number of species (elements) as their empirical counterparts, but they are species agnostic. In other words, instead of predicting a pair of values for each species directly (i.e. the number of prey and predators of a given species $i$), we predicted the entire joint degree sequence without taking into account species' identity (i.e. the distribution of the number of prey and predators for the entire set of species, without knowing which values belong to which species). The challenge is thus to adequately associate predictions with empirical data. In Fig 2, we present these differences when species are ordered by their total degree in their respective networks (i.e. by the sum of their in and out-degrees). This means that the species with the highest total degree in its network will be associated with the highest prediction, and so forth. Doing so, we see that species predicted to have a higher number of predators than what is observed generally have a lower number of prey than what is observed (and conversely). This is also shown in S1 Fig, which represents the relationship between prediction errors in the *absolute* (non-relative) values of $k_{out}$ and $k_{in}$ across networks of varying levels of species richness. This is because the difference in total degree ($k_{out} + k_{in}$) between predictions and empirical data is minimized when species are ranked by their total degree (i.e. the average deviation of the sum of relative $k_{out}$ and $k_{in}$ is close to 0 across all species). This result thus shows that the difference between predicted and empirical total degrees is low for most species when ordered by their total degrees. There are no apparent biases towards in or out degrees. In S2 Fig, we show how these differences change when species are instead ordered by their out-degrees (left panel) and in-degrees (right panel), i.e. when minimizing the error in the estimation of the out and in-degrees, respectively.

Another way to evaluate the empirical support of the sampled joint degree sequences is to compare their shape with the ones of empirical food webs. We described the shape of a joint degree sequence by measuring the distance between its in and out-degree sequences (i.e. the distance between its marginal distributions). To do so, we calculated the Kullback–Leibler (KL) divergence [48] between the in and out-degree sequences of each predicted and empirical distribution. The KL divergence is a measure of relative entropy describing the difference between two distributions. Low values indicate high similarity between the in and out-degree sequences and suggest that the joint degree sequence has a high level of symmetry. We compared the shape of the empirical and predicted joint degree sequences in the left panel of Fig 3.

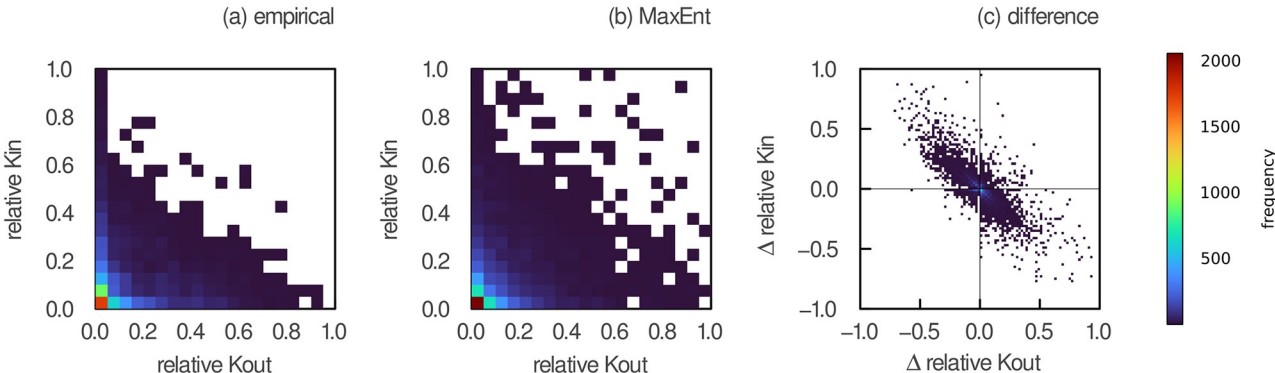

**Fig 2. Prediction errors of the relative number of predators and prey.** The relative number of predators ($k_{in}$) is plotted against the relative number of prey ($k_{out}$) for each species in all (a) empirical and (b) predicted joint degree sequences. The predicted joint degree sequences were obtained after sampling one realization of the joint degree distribution of maximum entropy for each network while keeping the total number of interactions constant. (c) Difference between predicted and empirical values when species are ordered according to their total degree. Due to significant data overlap, all relationships are represented as 2D histograms. The color bar indicates the number of species that fall within each bin.

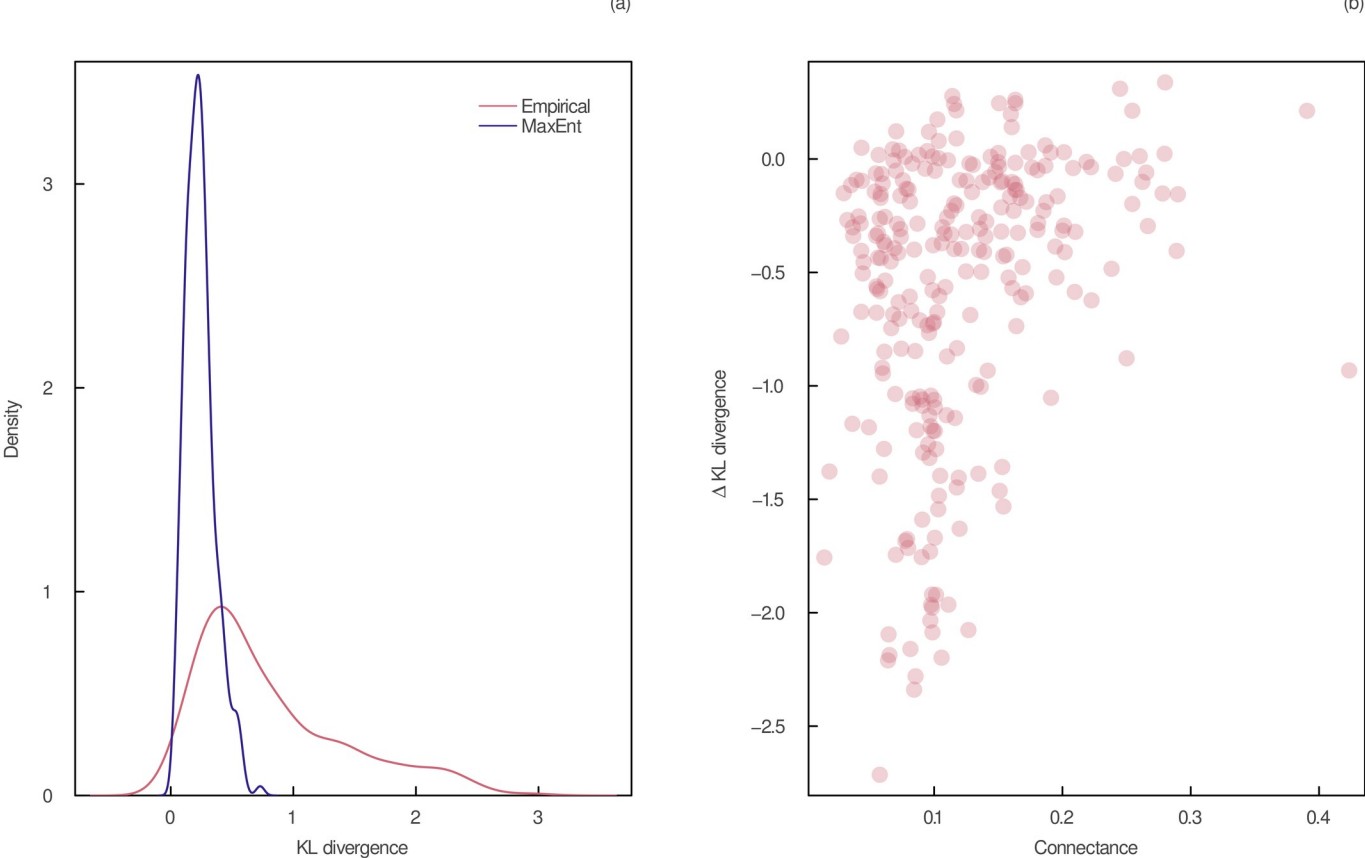

**Fig 3. Shape of empirical and predicted joint degree sequences.** a) Probability density of KL divergence between in and out-degree sequences of empirical and predicted joint degree sequences. (b) Difference between the KL divergence of empirical and predicted joint degree sequences as a function of connectance. The predicted joint degree sequences were obtained after sampling one realization of the joint degree distribution of maximum entropy for each network while keeping the total number of interactions constant.

As expected, our model predicts more similar in-degree and out-degree sequences than empirical data (shown by lower KL divergence values). However, the difference between the KL divergence of predicted and empirical joint degree sequences decreases with connectance (right panel of Fig 3). This might be because food webs with a low connectance are harder to predict than food webs with a high connectance. Indeed, in low connectance systems, what makes two species interact may be more important for prediction than in high connectance systems, in which what prevents species from interacting may be more meaningful. This implies that more ecological information may be needed in food webs with a low connectance because more ecological processes determine interactions compared to non-interactions. Therefore, other ecological constraints might be needed to account for the asymmetry of the joint degree distribution, especially for networks with a lower connectance. Nevertheless, our MaxEnt model seems to capture quite well the shape of the joint degree sequence for networks having a high connectance.

Regarding the degree distribution of maximum entropy, one aspect that informs us of its ecological realism is the number of isolated species it predicts. As [11] pointed out, the size of food webs should at least be of $S - 1$ interactions since a lower number would yield isolated species, i.e. species without any predator or prey. Because non-basal species must eat to

survive, isolated species could indicate that other species are missing; otherwise, isolated species should be removed from the network. In S3 Fig, we show that the degree distribution of maximum entropy, given $S$ and $L$, gives a very low probability that a species will be isolated in its food web (i.e. having $k = 0$) when $L > S - 1$. However, under our purely information-theoretic model, the probability that a species is isolated is quite high when the total number of interactions is below $S - 1$. Moreover, the expected proportion of isolated species rapidly declines by orders of magnitude with increasing numbers of species and interactions. This supports the ecological realism of the degree distribution of maximum entropy derived above. Nevertheless, ecologists wanting to model a system without allowing isolated species could simply change the lower limit of $k$ to 1 in Eq (16) and solve the resulting equation numerically.

## Heuristic maximum entropy models

In this section, we explore the predictions of our heuristic models. Overall, we found that the models based on the joint degree sequence (i.e. the type II null and heuristic MaxEnt models) reproduced the structure of empirical food webs much better than the ones based on connectance (i.e. the type I null and heuristic MaxEnt models, Table 1). This suggests that the predictive capacity of connectance might be more limited than what was previously suggested [10]. On the other hand, the neutral model of relative abundances was surprisingly good at predicting the maximum trophic level and the network diameter (Table 2). However, with the exception of the network diameter, the type II heuristic MaxEnt model was better at predicting

**Table 1. Standardized mean difference between predicted network measures and empirical data for all food webs in our complete dataset ($N$ = 257).**

| model | $\rho$ | maxtl | diam | MxSim | Cannib | Omniv | entropy |
|---|---|---|---|---|---|---|---|
| null 1 | -0.167 | 0.980 | 1.428 | -0.502 | 2.007 | 1.493 | 0.056 |
| MaxEnt 1 | -0.226 | 0.831 | 1.274 | -0.524 | 1.982 | 1.863 | 0.106 |
| null 2 | 0.160 | -0.125 | 0.016 | 0.007 | 1.078 | 0.559 | -0.023 |
| MaxEnt 2 | -0.015 | 0.178 | 0.565 | -0.282 | 0.698 | 0.589 | 0.058 |

Positive (negative) values indicate that the measure is overestimated (underestimated) on average. Null 1: Type I null model based on connectance. MaxEnt 1: Type I heuristic MaxEnt model based on connectance. Null 2: Type II null model based on the joint degree sequence. MaxEnt 2: Type II heuristic MaxEnt model based on the joint degree sequence. $\rho$: nestedness measured by the spectral radius of the adjacency matrix. *maxtl*: maximum trophic level. *diam*: network diameter. *MxSim*: average maximum similarity between species pairs. *Cannib*: proportion of cannibal species (self-loops). *Omniv*: proportion of omnivorous species. *entropy*: SVD entropy.

**Table 2. Standardized mean difference between predicted network measures and empirical data for all food webs in our abundance dataset ($N$ = 19).**

| model | $\rho$ | maxtl | diam | MxSim | Cannib | Omniv | entropy |
|---|---|---|---|---|---|---|---|
| neutral | 0.367 | -0.090 | 0.027 | 0.266 | 6.870 | 0.576 | -0.083 |
| null 1 | -0.134 | 0.950 | 1.919 | -0.369 | 2.077 | 0.614 | 0.068 |
| MaxEnt 1 | -0.229 | 1.020 | 1.946 | -0.355 | 2.215 | 0.801 | 0.121 |
| null 2 | 0.128 | -0.115 | -0.135 | 0.157 | 1.444 | 0.029 | -0.021 |
| MaxEnt 2 | -0.010 | 0.054 | 0.243 | -0.062 | -0.038 | 0.083 | 0.038 |

Positive (negative) values indicate that the measure is overestimated (underestimated) on average. Neutral: Neutral model of relative abundances. Null 1: Type I null model based on connectance. MaxEnt 1: Type I heuristic MaxEnt model based on connectance. Null 2: Type II null model based on the joint degree sequence. MaxEnt 2: Type II heuristic MaxEnt model based on the joint degree sequence. $\rho$: nestedness measured by the spectral radius of the adjacency matrix. *maxtl*: maximum trophic level. *diam*: network diameter. *MxSim*: average maximum similarity between species pairs. *Cannib*: proportion of cannibal species (self-loops). *Omniv*: proportion of omnivorous species. *entropy*: SVD entropy.

network structure than the neutral model for most measures considered. This might be because, although neutral processes are important, they act in concert with niche processes in determining species interactions [49–52]. The joint degree sequence captures information on both neutral and niche processes because the number of prey and predators a species has is determined by its relative abundance and biological traits. These results thus show that having information on the number of prey and predators for each species substantially improves the prediction of food-web structure, both compared to models solely based on connectance and to the ones solely based on species relative abundances.

Next, the predictions of the type II heuristic MaxEnt model can be compared to its null model counterpart. On average, the type II heuristic MaxEnt model was better at predicting nestedness (0.62 ± 0.08) than its corresponding null model (0.73 ± 0.05; empirical networks: 0.63 ± 0.09) for networks in our complete dataset (Table 1). This might in part be due to the fact that nestedness was calculated using the spectral radius of the adjacency matrix, which directly leverages information on the network itself just like the heuristic MaxEnt model. The proportion of self-loops (cannibal species) was also better predicted by the type II heuristic MaxEnt model in comparison to the type II null model. However, the type II null model was better at predicting network diameter and average maximum similarity between species pairs, and predictions of the maximum trophic level and the proportion of omnivorous species were similar between both types of models. We believe that this is because increasing the complexity of a food web might increase its average and maximum food-chain lengths. In comparison, the null model was more stochastic and does not necessarily produce more complex food webs with longer food-chain lengths.

Moreover, we found that the entropy of empirical food webs was slightly lower than their maximum entropy when constrained by their joint degree sequence (S4 Fig). Empirical food webs had an SVD entropy of 0.89 ± 0.04, compared to an SVD entropy of 0.94 ± 0.03 for networks generated using the type II heuristic MaxEnt model. The relationship between the SVD entropy of empirical food webs and their maximum entropy is plotted in the last panel of Fig 4. The slight increase in entropy confirms that our method generated more complex networks. Even though we found that many measures of empirical networks are close to the ones of their maximum entropy configuration, the relatively low predictability of entropy itself may be indicative of additional constraints shaping food-web structure, especially for networks with low SVD entropy. Incorporating more constraints into the model could increase its capacity to generate networks with an adequate level of complexity, as shown by the decrease in predictive errors of entropy of the type II heuristic MaxEnt model compared to the one based on connectance (Table 1). Additionally, we found no clear relationship between the increase in SVD entropy and the number of species, the number of interactions, and connectance (S5 Fig). This suggests that our model captured the complexity of small and large networks on a similar level and that its capacity to reproduce food-web structure was unrelated to the order and size of the network. In other words, the gap in entropy between empirical food webs and their maximum entropy configuration may be the result of additional constraints that were not taken into account in the model, regardless of the number of species and the number of interactions.

A direct comparison of the structure of maximum entropy food webs constrained by the joint degree sequence with empirical data also supports the results depicted in Table 1. In Fig 4, we show how well empirical measures are predicted by the type II heuristic MaxEnt model. Following our previous results, we found that nestedness was very well predicted by our model. However, the model overestimated the maximum trophic level and network diameter, especially when the sampled food web had intermediate values of these measures. In S6 Fig, we show that the pairwise relationships between the four measures in Fig 4 and species

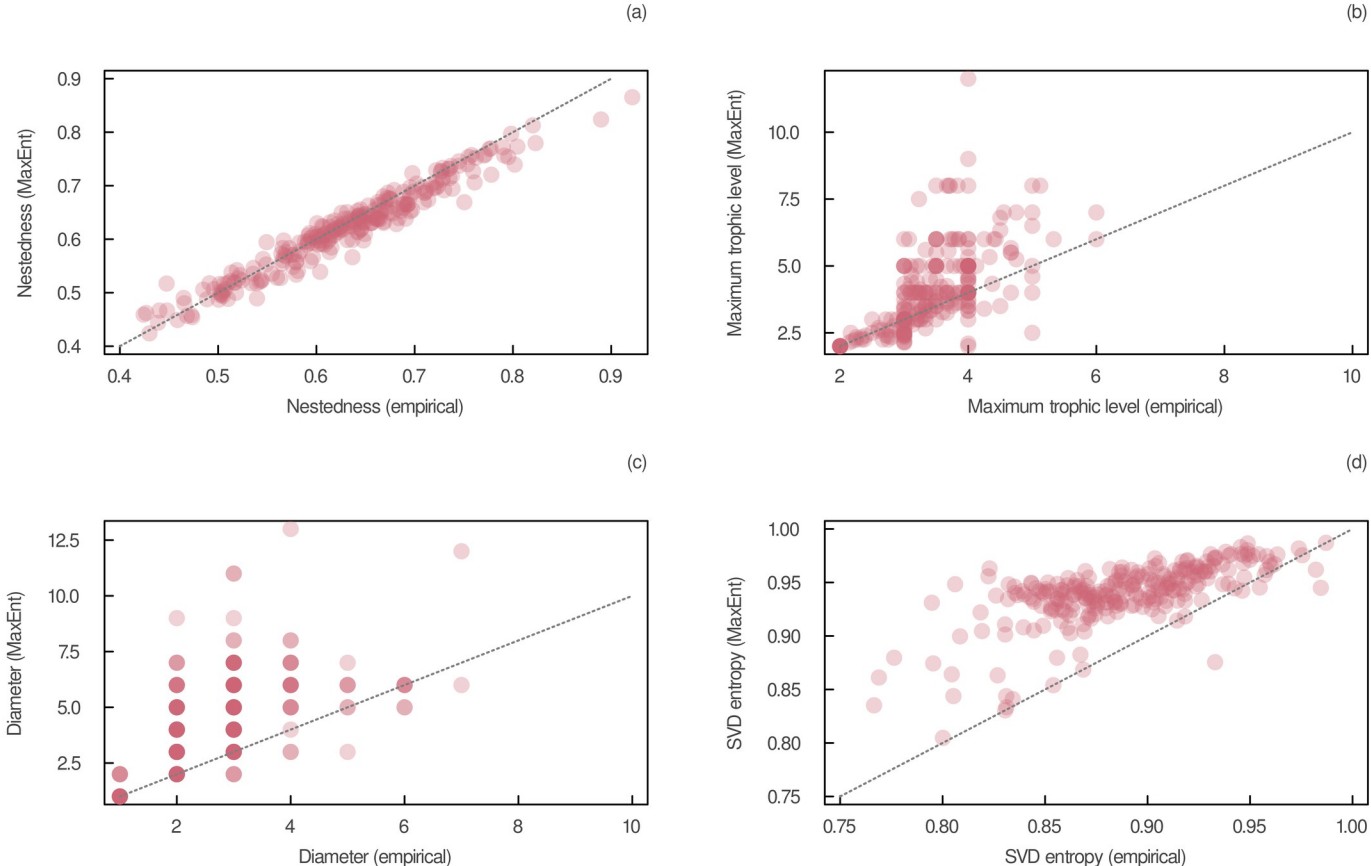

**Fig 4. Relationship between the structure of empirical and maximum entropy food webs.** Maximum entropy networks were obtained using the type II heuristic MaxEnt model based on the joint degree sequence. (a) Nestedness (estimated using the spectral radius of the adjacency matrix), (b) the maximum trophic level, (c) the network diameter, and (d) the SVD entropy were measured on these empirical and maximum entropy food webs. The identity line is plotted in each panel.

richness in empirical food webs are similar (in magnitude and sign) to the ones found in food webs generated using the type II heuristic MaxEnt model. This indicates that the number of species in the network does not seem to impact the ability of the model to reproduce food-web structure.

Notwithstanding its difficulties in reproducing adequate measures of food-chain lengths, the type II heuristic MaxEnt model can predict surprisingly well the proportions of three-species motifs in empirical food webs. Motifs have been shown to be the backbone of complex ecological networks on which network structure is built and play a crucial role in community dynamics and assembly [53]. Differences in motif profiles between an observed food web and null model-generated ones can unveil important ecological mechanisms that contribute to network structure [46]. In Fig 5, we show that the motif profile of networks generated using the type II heuristic MaxEnt model accurately reproduced the one of empirical data. This model made significantly better predictions than the ones based on connectance and the type II null model based on the joint degree sequence. This is also shown in Fig 6, which reveals that the relationships between the proportions of single-link motifs in empirical food webs are similar to the ones in networks generated using the type II heuristic MaxEnt model. This is in contrast with the type I null and MaxEnt models based on connectance, which produced opposite

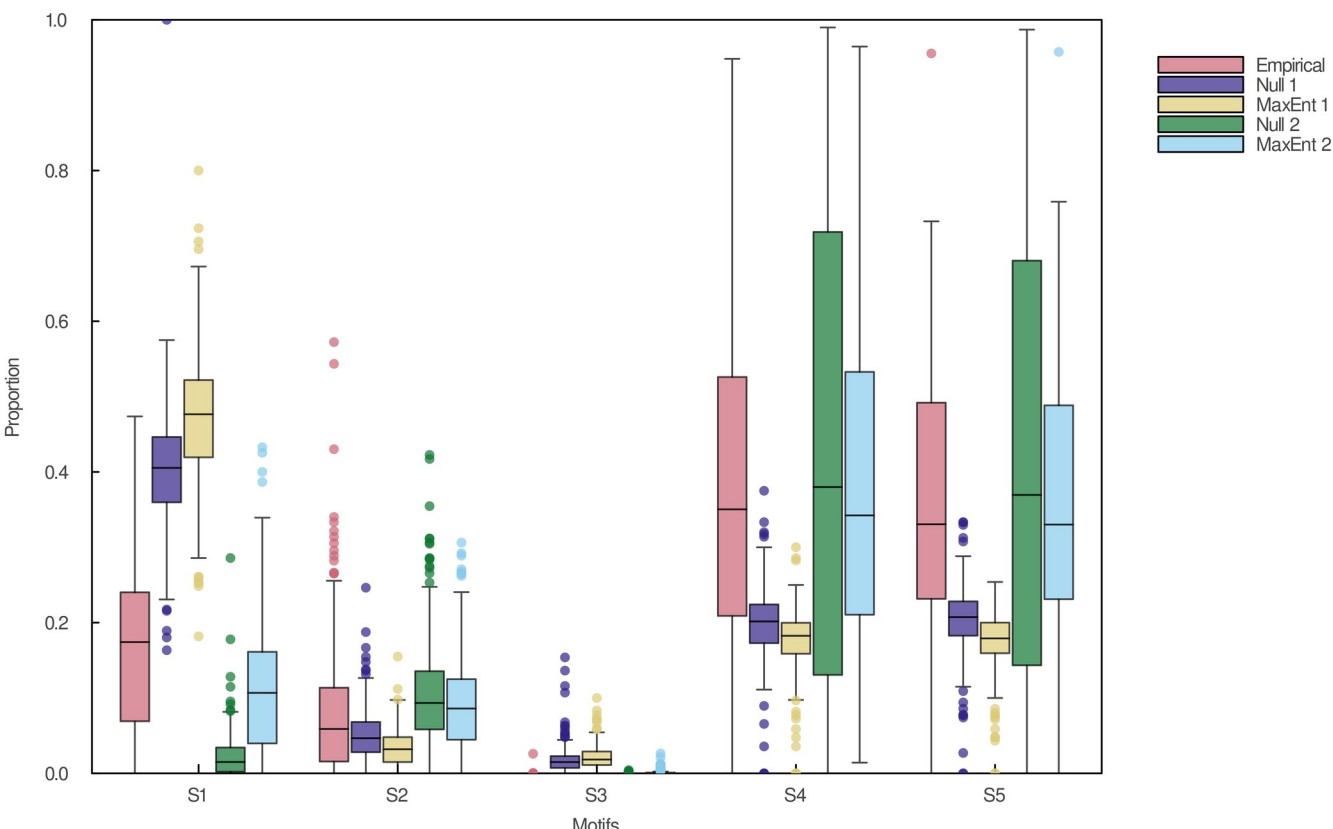

**Fig 5. Proportions of single-link three-species motifs in empirical and predicted food webs.** S1: Tri-trophic chain (a top predator feeds on a meso-predator which feeds on a basal prey). S2: Omnivory (a top predator feeds on a meso-predator and a basal prey). S3: Tri-trophic feeding loop (a cyclic three-species predator-prey system). S4: Apparent competition (a predator feeds on two prey). S5: Exploitative competition (two predators feed on the same prey). Null 1: Type I null model based on connectance. MaxEnt 1: Type I heuristic MaxEnt model based on connectance. Null 2: Type II null model based on the joint degree sequence. MaxEnt 2: Type II heuristic MaxEnt model based on the joint degree sequence. Boxplots display the median proportion of each motif in food webs (middle horizontal lines), as well as the first (bottom horizontal lines) and third (top horizontal lines) quartiles. Vertical lines encompass all data points that fall within 1.5 times the interquartile range from both quartiles, and dots are data points that fall outside this range. Only the single-link motifs S1-S5 are shown given the scarcity of double-link motifs in most empirical and predicted networks.

relationships than what was observed empirically. Our findings show that generating the most complex food web constrained by the joint degree sequence using maximum entropy does not alter the proportions of three-species motifs on the whole. This suggests that motif profiles may simply be a statistical attribute of food webs driven by the joint degree sequence. However, given the incapacity of our MaxEnt models to accurately predict food-chain lengths, the way motifs interconnect with each other may hold greater biological significance than the proportion of motifs itself.

One of the challenges in implementing and validating a maximum entropy model is to discover where its predictions break down. The results depicted in Table 1 and Fig 4 show that our type II heuristic MaxEnt model can capture many high-level properties of food webs, but does a poor job of capturing others. This suggests that, although the joint degree sequence is an important driver of food-web structure, other ecological constraints might be needed to account for some emerging food-web properties, especially the ones regarding food-chain lengths. Nevertheless, Figs 5 and 6 show that this model can reproduce surprisingly well motif profiles, one of the most ecologically informative properties of food webs. This suggests that the emerging structure of food webs is mainly driven by their joint degree sequence, although

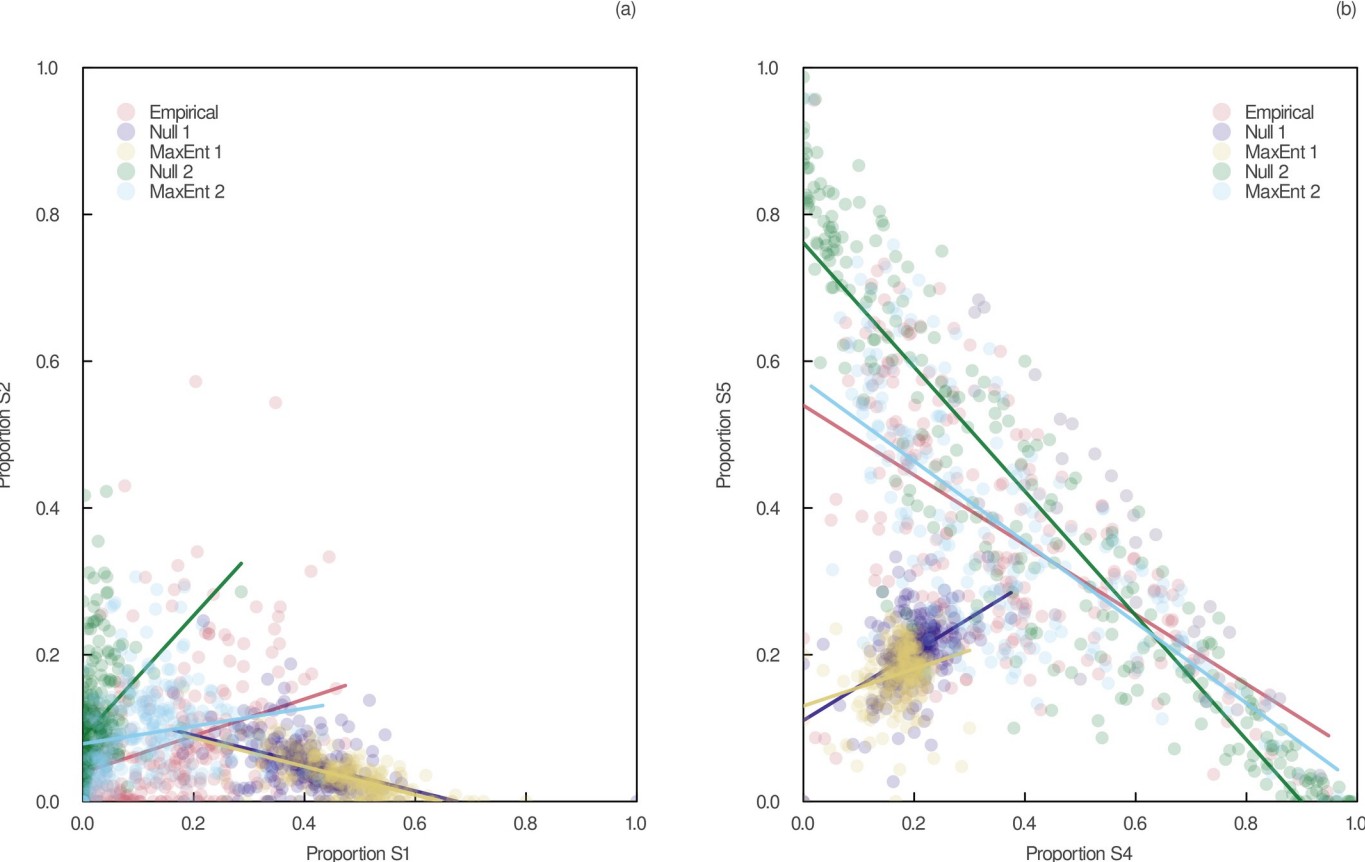

**Fig 6. Pairwise relationships between the proportions of single-link three-species motifs in empirical and predicted food webs.** S1: Tri-trophic chain. S2: Omnivory. S4: Apparent competition. S5: Exploitative competition. Null 1: Type I null model based on connectance. MaxEnt 1: Type I heuristic MaxEnt model based on connectance. Null 2: Type II null model based on the joint degree sequence. MaxEnt 2: Type II heuristic MaxEnt model based on the joint degree sequence. Regression lines are plotted in each panel. Motif S3 is not shown because of its low proportion in most empirical and predicted networks.

higher-level properties might need to be included in the model to ensure that food-chain lengths fall within realistic values.

## Conclusion

The principle of maximum entropy is a mathematical method of finding least-biased probability distributions that have some specified properties given by prior knowledge about a system. We first applied this conventional MaxEnt approach on food webs to predict species-level properties, namely the joint degree distribution and the degree distribution of maximum entropy given known numbers of species and interactions. We found that the joint degree distributions of maximum entropy had a similar shape to the ones of empirical food webs in high-connectance systems. However, these MaxEnt distributions were more symmetric than the ones of empirical food webs when connectance was low, which suggests that other constraints might be needed to improve these predictions in low-connectance systems. Then, we used a slightly different approach that aimed at finding heuristically the network configuration with the highest SVD entropy, i.e. whose vector of relative singular values has maximum entropy. This network of maximum entropy is the most complex, or random, given the specified structure. We found that the heuristic maximum entropy model based on connectance

did not predict the structure of sampled food webs very well. However, the heuristic maximum entropy model based on the entire joint degree sequence, i.e. on the number of prey and predators for each species, gave more convincing results. Indeed, this model reproduced food-web structure surprisingly well, including the highly informative motif profile. Nevertheless, it was not able to predict realistic food-chain lengths.

Our results bring to the forefront the role of the joint degree distribution in shaping food-web structure. This echoes the work of [9], who found that the degree distribution of ecological networks drives their emerging structure such as their nestedness and modularity. Several measures of food webs have been analyzed when studying the ecological consequences of network structure [1, 47]. In fact, following [30], we believe that there is a lot more ecological information in the deviation between these properties in empirical systems and their maximum entropy configuration given a fixed joint degree sequence.

## Alternative MaxEnt models

In this contribution, we used a method based on simulated annealing to find the network configuration with the highest SVD entropy while fixing some aspects of its structure. However, there are different ways to generate adjacency matrices using MaxEnt. Another technique, also based on simulated annealing, could begin by generating a food web randomly with fixed numbers of species and interactions and calculating its joint degree distribution. Pairs of interactions could then be swapped sequentially until we minimize the divergence between the calculated joint degree distribution and the one of maximum entropy obtained analytically. In that case, this is the entropy of the joint degree distribution that would be maximized, not the one of the network's topology. To a certain extent, this method would bridge the gap between the analytical and heuristic approaches presented in this article. More research is needed to compare the quality of different methods in generating adjacency matrices of food webs using MaxEnt.

Maximum entropy graph models are another type of method that predicts a distribution of adjacency matrices under soft or hard constraints (e.g. [25, 54]). Under hard constraints, every network with a non-zero probability exactly satisfies the constraints on its structure. This is in contrast with soft constraints, which require that networks satisfy them on average (i.e. many networks with a non-zero probability do not have the exact structure set by the constraints). Maximum entropy graph models are helpful because they can provide probability distributions for many network properties by measuring the structure of all adjacency matrices with a non-zero probability. However, we consider that our approach based on simulated annealing is more flexible and more computationally efficient. Indeed, many measures of food-web structure are hard to translate into mathematical constraints. Moreover, because food webs are directed networks that can have self-loops, it makes the mathematical derivation of maximum entropy graph models difficult. We believe that identifying heuristically what constrains the topology of food webs is a useful first step before attempting to derive the mathematical formulation of a maximum entropy graph model for food webs.

## Applications

Our analytical and heuristic models can be applied for different purposes. First, they could be used to generate first-order approximations of a network's properties when state variables are known empirically. For example, knowing the number of species in an ecological community, we can predict its number of interactions using the flexible links model and then predict its joint degree distribution with minimal biases using the principle of maximum entropy. This could prove particularly useful when predicting network structure at large spatial scales,

subdividing the study area into smaller communities (e.g. grid cells). Indeed, because species richness and other ecological data are increasingly abundant (e.g. [55]), validated MaxEnt models can be used to respond to a wider range of macroecological questions regarding food webs.

Second, our analytical model can be used to generate informative priors in Bayesian analyses of the structure of ecological networks (e.g. [56]). Indeed, the probability distribution of maximum entropy derived using MaxEnt can be used as a prior that can be updated with novel data. For instance, if we know the number of species and the number of interactions, we can derive the degree distribution of maximum entropy, as shown in this contribution. The degree distribution represents the probability that a species can interact (as a predator or a prey) with a given number of other species. Data on species interactions can be used to update the prior degree distribution to generate a more accurate posterior distribution, thus improving our description and understanding of the system.

Third, our analytical and heuristic models can be used to make better predictions of pairwise species interactions by constraining the space of feasible networks, as discussed in [14]. In other words, we can use the network configuration or specific measures of food-web structure derived using MaxEnt to ensure that our predictions of interspecific interactions form feasible networks. This means that the probability that two species interact may be conditional on the network structure and the probability of interactions of all other species pairs. When data are limited, MaxEnt can be used to predict network structure on which pairwise probabilities of interactions are conditional.

Finally, our analytical and heuristic models can be used as alternative null models of ecological networks to better understand and identify the ecological processes driving food-web structure. Indeed, these mechanisms can be better described when analyzing the deviation of empirical data from MaxEnt predictions [29]. A strong deviation would indicate that ecological mechanisms not captured by the statistical constraints are at play for the system at hand. For instance, the incapacity of a MaxEnt model to reliably predict entropy may be a compelling indication of additional constraints shaping food-web structure. If deviations are systematic, the maximum entropy model might need to be revised to include appropriate ecological constraints. This revision process helps us reflect on and identify what constrains food-web structure. However, it is important to note that tangible ecological mechanisms cannot be directly inferred from statistical distributions [57]. Instead, by identifying the constraints of a system and by analyzing empirical deviations from maximum entropy predictions, MaxEnt can only help us redirect research efforts toward understanding the biological mechanisms behind these constraints.

The principle of maximum entropy can thus be applied to both the prediction and understanding of natural systems. The model's interpretation depends on how we use it. It can be used as a baseline distribution to identify the ecological processes organizing natural systems. It can also be used to generate predictions of ecological networks. This distinction between understanding and predicting is important when using and interpreting MaxEnt models.

## Final remarks

One of the biggest challenges in using the principle of maximum entropy is to identify the set of state variables that best reproduce empirical data. We found that the number of species and the number of interactions are important state variables for the prediction of the joint degree distribution. Similarly, we found that the numbers of prey and predators for each species in a food web are important state variables for the prediction of the network configuration. However, our predictions overestimated the symmetry of the joint degree distribution for our

analytical model and the maximum trophic level and network diameter for our heuristic model. We should thus continue to play the ecological detective to find these other topological constraints that would improve the predictions of MaxEnt models and help us understand better what drives food-web structure.

## Supporting information

**S1 Fig. Prediction errors of the absolute number of predators $k_{in}$ and prey $k_{out}$.** Species were ordered according to their total degree in their network. Networks were sorted into different groups based on their total number of species. In each panel, each dot corresponds to a single species within one of the networks whose total species count is within the specified range. The predicted joint degree sequences were obtained after sampling one realization of the joint degree distribution of maximum entropy for each network while keeping the total number of interactions constant.
(TIF)

**S2 Fig. Prediction errors of the relative number of predators $k_{in}$ and prey $k_{out}$.** Species were ordered according to (a) their out-degree and (b) their in-degree. The predicted joint degree sequences were obtained after sampling one realization of the joint degree distribution of maximum entropy for each network while keeping the total number of interactions constant. Due to significant data overlap, all relationships are represented as 2D histograms. The color bar indicates the number of species that fall within each bin.
(TIF)

**S3 Fig. Probability that a species is isolated in its food web.** We derived many degree distributions of maximum entropy given a range of values of $S$ and $L$ and plotted the probability that a species has a degree $k$ of 0 (log-scale color bar). Species richness varies between 5 and 100 species, by increment of 5 species. For each level of species richness, the numbers of interactions correspond to all 20-quantiles of the interval between 0 and $S^2$. The black line marks the $S - 1$ minimum number of interactions required to have no isolated species.
(TIF)

**S4 Fig. SVD entropy of empirical and predicted food webs.** (a) Distribution of the SVD entropy of empirical and maximum entropy food webs. Maximum entropy networks were obtained using the type II heuristic MaxEnt model based on the joint degree sequence. (b) Distribution of z-scores of the SVD entropy of all empirical food webs. Z-scores were computed using the mean and standard deviation of the distribution of SVD entropy of MaxEnt food webs (type II heuristic MaxEnt model). The dashed line corresponds to the median z-score.
(TIF)

**S5 Fig. Prediction errors of SVD entropy.** Difference in SVD entropy between maximum entropy and empirical food webs as a function of (a) the number of interactions, (b) connectance, and (c) species richness. (d) Standardization of the difference in SVD entropy with respect to species richness as a function of species richness. The exponential decrease in the difference of SVD entropy per species with species richness offers a complementary perspective supporting the lack of relationship depicted in panel c. Maximum entropy networks were obtained using the type II heuristic MaxEnt model based on the joint degree sequence.
(TIF)

**S6 Fig. Structure of empirical and maximum entropy food webs as a function of species richness.** Maximum entropy networks were obtained using the type II heuristic MaxEnt model based on the joint degree sequence. (a) Nestedness (estimated using the spectral radius

of the adjacency matrix), (b) the maximum trophic level, (c) the network diameter, and (d) the SVD entropy were measured on these empirical and maximum entropy food webs and plotted against species richness. Regression lines are plotted in each panel.
(TIF)

## Acknowledgments

We acknowledge that this study was conducted on land within the traditional unceded territory of the Saint Lawrence Iroquoian, Anishinabewaki, Mohawk, Huron-Wendat, and Omàmiwininiwak nations.

## Author Contributions

**Conceptualization:** Francis Banville, Dominique Gravel, Timothée Poisot.

**Data curation:** Francis Banville, Timothée Poisot.

**Formal analysis:** Francis Banville, Dominique Gravel, Timothée Poisot.

**Funding acquisition:** Dominique Gravel, Timothée Poisot.

**Methodology:** Francis Banville, Dominique Gravel, Timothée Poisot.

**Project administration:** Dominique Gravel, Timothée Poisot.

**Software:** Francis Banville, Timothée Poisot.

**Supervision:** Dominique Gravel, Timothée Poisot.

**Validation:** Francis Banville, Dominique Gravel, Timothée Poisot.

**Visualization:** Francis Banville, Dominique Gravel, Timothée Poisot.

**Writing – original draft:** Francis Banville.

**Writing – review & editing:** Francis Banville, Dominique Gravel, Timothée Poisot.

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
