## [Decision Letter · Decision Letter 0]

19 May 2023

Dear PhD candidate Banville,

Thank you very much for submitting your manuscript "What constrains food webs? A maximum entropy framework for predicting their structure with minimal biases" for consideration at PLOS Computational Biology.

As with all papers reviewed by the journal, your manuscript was reviewed by members of the editorial board and by several independent reviewers. In light of the reviews (below this email), we would like to invite the resubmission of a significantly-revised version that takes into account the reviewers' comments.

While I have chosen Major revisions as the category, qualitatively, I would put this somewhere in between Major and Minor revisions. I believe the authors can successfully comment on the reviewrs queries and find the suggestions helpful in improving the readability of the manuscript.

We cannot make any decision about publication until we have seen the revised manuscript and your response to the reviewers' comments. Your revised manuscript is also likely to be sent to reviewers for further evaluation.

Sincerely,

Chaitanya S Gokhale, PhD

Academic Editor

PLOS Computational Biology

Natalia Komarova

Section Editor

PLOS Computational Biology

Reviewer's Responses to Questions

**Comments to the Authors:**

Reviewer #1: The manuscript “What constrains food webs? A maximum entropy framework for predicting their structure with minimal biases” presents an elegant strategy to develop null models to describe food networks. The advantage of the proposed strategy, relying on the maximum entropy framework, is its simplicity and versatility, nicely boosted by the possibility of using only S, the number of species, to generate null predictions about several features of the networks. For food networks, null models already exist, but the ones presented in this manuscript seem to capture better the properties of empirical food networks; in addition, I’d argue that the maximum entropy framework presents a more solid foundation for such null models. Finally, the paper describes the main framework very clearly, and it is well written. Thus, I am sure it will allow ecologists not versed in information theory to implement such null models easily. Therefore, I welcome this paper and I recommend its publication in the present form, with only minor adjustments. I only have a few minor comments, mostly related to the form and the comparisons with the empirical data, which I think could be improved.

Motif profiles are not described at all in the text, nor in the figure caption. The authors refer to Stouffer 2007, but I think that adding a minimal description would significantly improve readability. This is important because I think some of the strongest points in using MaxEnt2 and using joint degree sequence as constraint is made by figure 5 and 6. It is nice how MaxEnt2 seems to predict better and more consistently all motif profiles. However, at the moment, these are not completely transparent. For instance, in figure 6 I notice that motif S3 is not shown. Is that because of the low total abundance?

Figure 1 could be improved. There is a lot of overlap of points, maybe a heatmap or a different representation could be better. Also the layout, spacing and so on could be improved. I also understand why normalizing relative Kout and Kin, but perhaps there is enough space (if not, in the supplementary) to stratify by connectance or S, particularly figure c.

The sentence “Empirical networks include most food webs archived on Mangal, as well as the New Zealand and Tuesday Lake datasets (our complete dataset)” is repeated over and over in every caption. I don’t think it is needed since all analyses are done on that dataset. Expect table 2?

Weird phrasing in line 416 (“We found no to a weak..”). I think it would also be nice if the authors could comment more on Fig.S4, which seems to show the fact that SVD Entropy is more poorly predicted for larger networks. Perhaps one should standardize this somehow?

I also think it is quite nice that the authors show that their null models predict very well several features of food webs, motif profiles, nestedness, etc. Given this result, it is quite interesting that instead entropy SVD-entropy itself is less well predictable, especially since it is a at the core of the model. I wonder if this means that motif profiles as currently defined are not that biologically relevant, and are merely a statistical property of these networks. Could the authors reflect on this, and maybe add a line or two in the discussion about potential applications.

Reviewer #2: What constrains food webs? A maximum entropy framework for predicting their structure with minimal biases

The article delves into a crucial topic in ecology, which is essential to comprehend and predicting ecological networks. It thoroughly discusses MaxEnt, a model that proves to be a valuable tool for ecologists, allowing researchers to understand connectance and ecosystem structure better. The authors explain the applications, motivations, and mathematical formulas clearly and concisely. Their findings are significant and present an exciting approach to their use in real-world data, hence having great applicability in conservation. However, the article could benefit from more elaboration on technical terms to help non-specialised readers.

Introduction

9-12. The knowledge gaps surrounding web properties need to be addressed. The author's reasoning for the complexity of ecological network structure and behaviour needs to be clarified. Consider rephrasing the sentence.

16-17. Define nodes and edges in an earlier moment.

17. Remove "indeed."

17-22. The sentences seem disconnected. Rephrase.

24: Define "maximum trophic level" earlier in the text.

25. Remove "thus".

26. Remove "indeed."

26. Define "Eltonian shortfall."

28. Please expand on how network constraints could be used to predict pairwise interactions and network properties. Especially they might be relevant to the maximum entropy framework.

34. Remove "thus".

34-35. The text needs to clarify the distinction between understanding and predicting.

36. Define "null models" before discussing their benefits.

37-38. Needs clarification.

40. Define "predictive models".

52-52. Please expand on this.

56-57. Is there a particular reason on why entropy would have a different definition from physics (i.e. uncertainty of a random process)? If so, the reader might benefit from knowing, especially when a widely known term is redefined.

61. Is "discrete case" an essential aspect of this sentence? Discrete can be a data property, and it might confuse readers. Rephase to " a measure of entropy should satisfy four properties:"...

84. Nodes and edges should have been defined earlier in the text.

103-104. Clarify the choice to use deterministic and unweighted food webs.

107-110. Needs clarification.

114. Needs citation.

137-148. In complex food webs, such as the data employed, joint probability distribution is often miscalculated given the number of interactive components in the system. How does this noise affect MaxEnt estimates?

198-199. Can you expand on joint degree measurements versus individual measurements? Also, if an element (predator or prey) is present in multiple sequences, would this be accounted for repeatedly? If so, would this affect the prediction estimates of a network?

213-216. Needs clarification.

230. Remove "indeed"

263. Why a beta-binomial distribution? Stable interaction webs usually have an exponential distribution. Clarify if the distribution choice is due to the results in figure 3.

267. Monte Carlo simulation describes an arbitrary interaction process. Would the algorithm choice influence the determination of a non-arbitrary process, such as food networks?

268. Why a single chain? Simulations usually have 2+ to ensure proper mixing.

310. Remove "indeed"

316. Define the acronym "SVD" in an earlier moment.

317-329. That is a great definition of the MaxEnt process. The authors might consider moving this session to when MaxEnt was first introduced.

366. Define "motifs"

333. Define SVD.

421. Remove "indeed"

Reviewer #3: Summary: I believe I was asked to review this paper because of my knowledge of food web models and of mathematical modeling in ecology, so I will restrict my comments to these specific themes. Briefly, it is my opinion that this paper should be published with minor revisions. It adds to a growing body of literature on MaxEnt work within ecology and makes important contributions towards predicting food web structure.

Overview of article’s findings/my understanding of the findings: The authors leverage the principal of maximum entropy to find the least-biased probability distribution given the constraints of a food web. This least-biased probability distribution is the one with the highest entropy among all probability distributions that satisfy the given constraints. The authors chose to use two complementary approaches to predict the structures of food webs using the maximum entropy principle. The first approach consists of deriving constrained probability distributions of given network properties analytically, while the second approach consists in finding the adjacency matrix of maximum entropy heuristically. These attempts at predicting food web structures through the principle of maximum entropy are novel and build the groundwork for the use of MaxEnt in food web studies. The authors found mixed results when attempting to predict food web structures with MaxEnt. They found that MaxEnt predicts the joint degree distributions well in high-connectance systems, but when connectance was low MaxEnt distributions were too symmetrical. In their heuristic approach, they found that MaxEnt models based on connectance did not predict food web structure very well. They did, however, find that the heuristic model based on joint degree distributions reproduced food web structure quite well. Nevertheless, this approach was still not able to predict realistic food-chain lengths.

Recommendation: The authors do a good job grounding this study in MaxEnt, network, and food web literature. They complete multiple tests on both of their approaches, running their MaxEnt models against null models, neutral models, and against empirical food webs. In novel research like this, running many tests is important to show where this strategy works and where it does not. While their results show that MaxEnt may not be ideal for certain predictions of food webs with specific constraints such as low-connectance, they did find that in high-connectance systems the analytical method works well for predicting joint degree distributions. This paper builds the basis for the application of the principle of maximum entropy for predicting food webs. I believe this is an important contribution to the future of food web studies, both the successes and failures of their MaxEnt predictions.

Overall, this paper is well written and shows how novel research may not produce expected outcomes, but that certain aspects can be used for future applications. This paper makes an impact in the way that it lays the foundation to use MaxEnt principles for structural predictions of food webs.

I recommend publication with (very) minor revisions.

Suggested changes: I suggest that on line 263 when discussing the beta-binomial distribution that the authors make note that “BB” in the following two equations (16 & 17) represent the beta-binomial distribution. That could be represented in the sentence as such, “…obtained from a beta-binomial distribution (BB)...” just to clarify for the readers who may not be familiar with beta-binomial distributions. I recognize that “BB” is a standardized scientific nomenclature, however, it would be helpful to explain it in its first usage. The details of the methodology are, as mentioned, easy enough to follow which allows for this project to be replicated.

I might also suggest a glance at Caruso et al. (2022) “Fluctuating ecological networks: A synthesis of maximum-entropy approaches for pattern detection and process inference” as a paper to cite when referencing MaxEnt approaches in ecology. This is another paper that analyzes the ability to use MaxEnt with the constraints of ecological networks. This paper does not look specifically at food webs; however, it does discuss the applications of MaxEnt in ecology. So, it might be pertinent to include this new research in your citation list.

Finally, I wanted to layout some of the minutia that should be fixed before publication:

• Line 254: “However, since the later is rarely measured empirically…” I believe this should be “latter” instead of “later”.

• Line 309: “This type of models…” I believe models should either be singular or “this type” should be plural.

Specific comments required by PLoS:

The authors meticulously broke down their methodology and the equations associated with their predictions. I personally found it easy to follow their succession of equations, which build on one another throughout their methodology sections.

The authors’ inclusion of potential future applications of this work supports their findings and demonstrate this paper’s function as footing for using MaxEnt in food web studies. They noted where it works when predicting food webs and where it does not. This is important information when discussing potential future applications for MaxEnt in food webs research. The authors lay out four specific potential applications for MaxEnt in food web structural research. This paper tackles the important task of deciphering state variables that do and do not reproduce empirical data.

Has the author-generated code that underpins the findings been made publicly available?

The authors mention that all code and data for reproduction purposes are available at the Open Science Framework and were made in open-source coding language Julia. So, beyond the opportunities for future applications of this work, they have made it accessible for replication via open-source software.

**Have the authors made all data and (if applicable) computational code underlying the findings in their manuscript fully available?**

Reviewer #1: Yes

Reviewer #2: Yes

Reviewer #3: Yes

PLOS authors have the option to publish the peer review history of their article (what does this mean?). If published, this will include your full peer review and any attached files.

Reviewer #1: No

Reviewer #2: No

Reviewer #3: **Yes: **Evan A. Holt
---

## [Decision Letter · Decision Letter 1]

22 Aug 2023

Dear PhD candidate Banville,

We are pleased to inform you that your manuscript 'What constrains food webs? A maximum entropy framework for predicting their structure with minimal biases' has been provisionally accepted for publication in PLOS Computational Biology.

I would like to apologise for the delay in getting ot the decision but I wanted to give the reviewers the full chance to go thorugh the changes. Perhaps due to the summer holidays in the northern hemisphere, the process took longer than expected. I have gone through the changes that you have made and it reads really well! Congratulations!

Best regards,

Chaitanya S Gokhale, PhD

Academic Editor

PLOS Computational Biology

Natalia Komarova

Section Editor

PLOS Computational Biology

Reviewer's Responses to Questions

**Comments to the Authors:**

Reviewer #1: Thanks to the authors for having addressed all requests. I believe the paper is now ready for publication.

**Have the authors made all data and (if applicable) computational code underlying the findings in their manuscript fully available?**

Reviewer #1: Yes

PLOS authors have the option to publish the peer review history of their article (what does this mean?). If published, this will include your full peer review and any attached files.

Reviewer #1: No

---

## [Editor Report · Acceptance letter]

31 Aug 2023

PCOMPBIOL-D-23-00390R1 

What constrains food webs? A maximum entropy framework for predicting their structure with minimal biases

Dear Dr Banville,

I am pleased to inform you that your manuscript has been formally accepted for publication in PLOS Computational Biology. Your manuscript is now with our production department and you will be notified of the publication date in due course.

With kind regards,

Anita Estes
